# Lipid Profile Is Negatively Associated with Uremic Toxins in Patients with Kidney Failure—A Tri-National Cohort

**DOI:** 10.3390/toxins14060412

**Published:** 2022-06-16

**Authors:** Sam Hobson, Henriette de Loor, Karolina Kublickiene, Joachim Beige, Pieter Evenepoel, Peter Stenvinkel, Thomas Ebert

**Affiliations:** 1Division of Renal Medicine, Department of Clinical Science, Intervention and Technology, Karolinska Institutet, SE-17177 Stockholm, Sweden; karolina.kublickiene@ki.se (K.K.); peter.stenvinkel@ki.se (P.S.); 2Nephrology and Renal Transplantation Research Group, Department of Microbiology, Immunology and Transplantation, Katholieke Universiteit Leuven, BE-3000 Leuven, Belgium; jetty.deloor@uzleuven.be (H.d.L.); pieter.evenepoel@uzleuven.be (P.E.); 3Division of Nephrology and KfH Renal Unit, Hospital St. Georg, 04129 Leipzig, Germany; joachim.beige@kfh.de; 4Martin-Luther-University Halle-Wittenberg, 06108 Halle, Germany; 5Department of Nephrology and Renal Transplantation, University Hospitals Leuven, BE-3000 Leuven, Belgium; 6Medical Department III—Endocrinology, Nephrology, Rheumatology, University of Leipzig Medical Center, D-04103 Leipzig, Germany

**Keywords:** cholesterol, lipids, lipoproteins, renal disease, triglycerides, uremic retention solutes, uremic toxins

## Abstract

Patients with kidney failure (KF) have a high incidence of cardiovascular (CV) disease, partly driven by insufficient clearance of uremic toxins. Recent investigations have questioned the accepted effects of adverse lipid profile and CV risk in uremic patients. Therefore, we related a panel of uremic toxins previously associated with CV morbidity/mortality to a full lipid profile in a large, tri-national, cross-sectional cohort. Total, high-density lipoprotein (HDL), non-HDL, low-density lipoprotein (LDL), and remnant cholesterol, as well as triglyceride, levels were associated with five uremic toxins in a cohort of 611 adult KF patients with adjustment for clinically relevant covariates and other patient-level variables. Univariate analyses revealed negative correlations of total, non-HDL, and LDL cholesterol with all investigated uremic toxins. Multivariate linear regression analyses confirmed independent, negative associations of phenylacetylglutamine with total, non-HDL, and LDL cholesterol, while indole-3 acetic acid associated with non-HDL and LDL cholesterol. Furthermore, trimethylamine-N-Oxide was independently and negatively associated with non-HDL cholesterol. Sensitivity analyses largely confirmed findings in the entire cohort. In conclusion, significant inverse associations between lipid profile and distinct uremic toxins in KF highlight the complexity of the uremic milieu, suggesting that not all uremic toxin interactions with conventional CV risk markers may be pathogenic.

## 1. Introduction

Patients with chronic kidney disease (CKD) undergo premature ageing, which represents a discrepancy between chronological and biological age [1]. Furthermore, increased chronic inflammation, oxidative stress, and cellular senescence are frequently observed in CKD [2] contributing to increased mortality with 40–50% of deaths attributed to cardiovascular disease (CVD) [3]. Patients with CKD are especially prone to develop early vascular ageing, a sequelae characterized by changes in vascular structure and function, including but not limited to medial calcification, vascular smooth muscle cell migration/proliferation/differentiation and endothelial dysfunction, ultimately leading to increased vessel stiffness and a substantially higher cardiovascular (CV) risk [4,5]. Despite considerable progress in cardiometabolic research, the pathophysiological mechanisms of the CV burden in kidney failure (KF) driven by the toxic uremic milieu are not fully understood, thus therapeutic interventions are limited.

Uremic toxins are compounds which accumulate in patients as kidney function declines [6]. Uremic toxins have been classified into different groups based on their size and molecular weight, their pathophysiologic effect, and solute origin [7,8]. In recent years, the pathogenic roles of distinct uremic toxins and how they contribute to progress CKD have been elucidated. For example, protein-bound indoxyl sulphate has been shown to promote renal tubular cell toxicity (reviewed in [9]) and CV complications such as vascular calcification [10,11], endothelial dysfunction [12], and vascular smooth muscle cell proliferation [13], while low-molecular-weight solute trimethylamine-N-Oxide (TMAO) induces renal fibrosis [14], and has also been associated with CV complications [15].

Similarly, modifications in lipid metabolism are frequently observed in CKD patients, which can accelerate CKD progression, and promote existing and secondary CVD [16,17]. Thus, patients with KF show hypertriglyceridemia and decreased cardioprotective high density lipoprotein (HDL) cholesterol as compared to healthy individuals [16].

However, the interplay between uremic toxins and lipid profile is largely unstudied, although a growing body of evidence suggests that individual toxins regulate lipoprotein function. For example, the uremic toxin symmetric dimethylarginine was shown to accumulate in HDL cholesterol, modulating its function by inhibiting anti-inflammatory properties, and was associated with reduced reverse cholesterol efflux and increased mortality [18]. In accordance, our group recently demonstrated that lipid parameters are counterintuitively inversely associated with mortality in incident hemodialysis patients [19] suggesting a uremia-induced dysregulation of lipid profile.

To unravel the link between lipid profile and uremic toxins, we investigated the association between five uremic toxins with comprehensive evidence of promoting CVD as a single entity (i.e., indoxyl sulphate [11,20], p-cresyl sulphate [20], indole-3 acetic acid (IAA) [21], TMAO [22], and phenylacetylglutamine (PAG) [23]) and a full lipid profile (i.e., total, HDL, non-HDL, low density lipoprotein [LDL] and remnant cholesterol, as well as triglycerides) in a tri-national cohort of 611 KF patients from Sweden, Belgium and Germany.

## 2. Results

### 2.1. Baseline Characteristics

Baseline characteristics of the entire cohort, as well as after stratification based on sex and dialysis status, are shown in Table 1. Median [interquartile range] age of the entire cohort was 55 (43–67) years, and 69.2% were treated by dialysis therapy (Table 1). Presence of diabetes, statin usage, and dialysis vintage did not differ between sexes (all *p* > 0.05; Table 1). In contrast, female subjects had higher levels of total, HDL, non-HDL, and LDL cholesterol compared to male subjects (all *p* < 0.05; Table 1) but did not differ in circulating levels of remnant cholesterol or triglycerides (all *p* > 0.05; Table 1). Furthermore, serum creatinine was significantly higher in male subjects compared to females (*p* < 0.001; Table 1). Moreover, indoxyl sulphate serum levels were significantly higher in male (92.8 [54.1–143.9] µmol/L) compared to female (80.5 [31.4–126.6] µmol/L) subjects (*p* = 0.005; Table 1), whereas all other investigated uremic toxins did not differ between sex strata (all *p* > 0.05; Table 1). Patients on dialysis showed lower values for all lipid parameters (all *p* < 0.05; Table 1). In contrast, uremic toxin levels were increased in individuals on dialysis (all *p* < 0.05; Table 1).

### 2.2. Univariate Correlations of Lipid Profile and Uremic Toxins

By applying strict Bonferroni correction, lipid markers were correlated with the five investigated uremic toxins in the entire cohort. Total, non-HDL, and LDL cholesterol all showed a similar pattern of regulation, i.e., a negative and significant correlation with all uremic toxins (all *p* < 0.05, Figure 1, Appendix A). Furthermore, HDL cholesterol was significantly and negatively related to indoxyl sulphate (*p* < 0.05, Figure 1, Appendix A).

### 2.3. Multivariate Regression Analyses

To identify independent associations between uremic toxins and lipid profile, we performed multivariate linear regression analyses with adjustment for age, sex, cohort, body mass index (BMI), presence of diabetes, high sensitivity C-reactive protein (hsCRP), statin use, estimated glomerular filtration rate (eGFR), and dialysis treatment only for those parameters that were significantly correlated to lipid markers in univariate analyses. Multivariate analyses reveal that IAA was independently, negatively, and significantly related to non-HDL (standardized beta = −0.150) and LDL (standardized beta = −0.122) cholesterol (both *p* < 0.05; Table 2, Appendix A). Furthermore, TMAO was independently and negatively associated with non-HDL cholesterol (standardized beta = −0.079; *p* < 0.05; Table 2, Appendix A). Moreover, PAG was independently and negatively related to total (standardized beta = −0.095), non-HDL (standardized beta = −0.133), and LDL cholesterol (standardized beta = −0.071) (all *p* < 0.05; Table 2, Appendix A). In contrast, indoxyl sulphate and p-cresyl sulphate were not significantly and independently related to lipid markers after multiple adjustment (all *p* > 0.05; Table 2, Appendix A).

When eGFR was removed from the full multivariate model, the strength of all associations increased in terms of effect size (i.e., standardized beta), and total, non-HDL, and LDL cholesterol were independent, negative predictors of IAA, TMAO, and PAG (data not shown). Furthermore, when the analysis was not adjusted for eGFR, indoxyl sulphate was independently and negatively associated with total and HDL cholesterol (data not shown).

### 2.4. Sensitivity Analyses

We further re-analyzed our tri-national cohort stratified by study center, dialysis therapy, incident vs. prevalent dialysis therapy, and sex. First, we investigated all associations separately for each study center, i.e., Stockholm, Leuven, and Leipzig. Here, while all significant associations after adjustment in the full model (i.e., Table 2) remained negative in the stratified models, some did not reach statistical significance. For instance, PAG’s significant association with total, LDL, and non-HDL cholesterol was lost in the Stockholm subcohort, while significance was lost in all models in the Leuven cohort, and only two models in Leipzig cohort remained significant (between PAG and total/HDL cholesterol) (Appendix A).

Multivariate regression analyses stratified by dialysis therapy revealed that the negative associations between lipid profile and uremic toxins remained virtually the same (Appendix A) in terms of effect size as in the full models (Table 2), although statistical significance was lost for some of the associations (Appendix A). Thus, PAG was not significantly related to lipid markers in non-dialysis subjects after stratification. However, IAA was negatively associated with total cholesterol in non-dialysis patients, however this was not observed in models including all patients or the dialysis strata (Appendix A).

When a subcohort of patients treated with dialysis therapy was investigated separately based on incident vs. prevalent dialysis therapy, many of the independent associations lost significance in both strata (Appendix A). Interestingly, protein-bound p-cresyl sulphate was positively and independently related to total, non-HDL, and LDL cholesterol in incident, but not in prevalent, dialysis patients (Appendix A).

Importantly, when the multivariate model was re-analyzed in study participants from Stockholm and Leuven (N = 388) with further adjustment for serum albumin, the associations of uremic toxins and lipid profile remained virtually the same compared to the full model without albumin in terms of effect size (i.e., standardized beta coefficients), although some associations lost statistical significance (data not shown).

Finally, when stratified by sex, multivariate analyses showed overall comparable associations between lipid profile and uremic toxins (Appendix A) to those in the full model (Table 2). There was, however, a negative independent association between indoxyl sulphate and HDL cholesterol in male subjects, which was not observed in the full model or in female subjects (Appendix A).

## 3. Discussion

We investigated the association of lipid biomarkers with serum concentrations of five uremic toxins in a large, multicenter cohort study in patients with advanced CKD/KF. Our main finding is that multiple uremic toxins with proven or putative cardiovascular toxicity show a negative association with lipid biomarkers, which remains significant after adjustment for clinically relevant covariates.

Uremic toxins accumulating in advanced CKD and KF, including indoxyl sulphate [11], p-cresyl sulphate [24], and PAG [23], have been associated with poor prognosis and outcomes. Furthermore, uremic dyslipidemia is characterized by decreased levels of atheroprotective HDL cholesterol, hypertriglyceridemia, and increased LDL cholesterol oxidation, which in concert may have implications in promoting monocyte chemotactic activity and endothelial dysfunction [16,25]. Despite the undeniable harmful effects of uremic toxins and/or dyslipidemia in CKD on multiple organs in the body, only a few smaller studies have investigated links between uremic toxins as single entities and distinct lipid markers.

Using a panel of five clinically well-established uremic toxins in a large, tri-national, KF patient cohort, we demonstrate counter-intuitive findings: IAA, TMAO, and PAG all negatively associate with non-HDL cholesterol; IAA and PAG negatively associate with LDL cholesterol; and PAG is negatively related to total cholesterol, all after adjustment for clinically relevant covariates. Interestingly, IAA and PAG are most strongly related to reduced lipid levels. Although the study design does not allow us to determine causality, three explanations could explain the observed effects: (1) uremic toxins may affect lipid metabolism; (2) vice versa; and/or (3) common culprits may be involved.

It is tempting to speculate that uremic toxins, including indoxyl sulphate, IAA, TMAO, and PAG, mechanistically alter the uremic lipid profile. As an example, indoxyl sulphate was shown to directly stimulate oxidized LDL cholesterol uptake in macrophages [26] and reduce cholesterol efflux in macrophages in vitro [27]. The inverse association between IAA and cholesterol is in accordance with animal experiments showing that IAA treatment lowers plasma total and LDL cholesterol in high fat diet-fed mice [28]. This is further supported by the uremic milieu’s ability to increase molecular diversity of lipoproteins, as described by Noels et al. [16]. In more detail, molecular pathways by which uremic toxins might alter lipid profile in CKD and contribute to CVD include increased oxidative stress resulting in modified LDL cholesterol (e.g., oxidized LDL [29,30,31]), post-translational modifications of lipoproteins (e.g., carbamylated LDL [30,31]), lipoprotein changes through the direct induction of insulin resistance [30,31], as well as potentiating adverse CV effects of circulating lipid particles [30]. Therefore, based on the definition of a uremic toxin, an altered lipid profile may even constitute as a new class of uremic toxins [30,31].

To the best of our knowledge, a link between lipid metabolism and PAG has not been previously reported. In contrast, most published evidence suggest a direct stimulation of adverse cholesterol metabolism by the uremic toxin TMAO [32]. In accordance, inhibiting TMAO production by a knockdown of hepatic flavin-containing monooxygenase 3 entirely prevents the development of dyslipidemia and atherosclerosis in mice [33]. Importantly, these studies were performed in mice without evidence of renal dysfunction, and the presence of KF might explain the observed counter-intuitive data on TMAO in our cohorts.

On the contrary, our findings may also be explained by lipid metabolism having a direct effect on uremic toxins. We can also not rule out a common mechanism driving a reduction in lipid profile markers while simultaneously increasing uremic toxin concentrations: For example, a “western” high fat diet, which has been shown to induce microbial dysbiosis and alter gut barrier function (thus possibly uremic toxin generation [34]), may also have direct effects on specific components of the lipid profile [35]. Future prospective studies on the associations of uremic toxins with cardiometabolic risk markers, including lipid profile, should therefore always include dietary information of the included study participants.

In addition, reverse epidemiology, a unique CV risk factor profile in KF [25], cholesterol-mediated endotoxin binding [36], as well as inflammation and/or protein energy wasting [37], may also contribute to our observed findings. It is important to note that our associations remain statistically significant even after adjustment for markers of inflammation, statin treatment, as well as further CV risk factors. Furthermore, statin treatment does not affect uremic toxins’ independent significant associations with lipid profile (i.e., IAA, TMAO, and PAG) in our tri-national cohort. Moreover, the ARO hemodialysis study has recently demonstrated an inverse relationship between lipid profile and mortality independent of the aforementioned covariates [19].

In sensitivity analyses, we show only marginal differences in multivariate regression models when patients were stratified by study center, dialysis treatment, dialysis duration, and sex compared to the full cohort multivariate regression models. We speculate that incompletely understood sex differences between subjects and uremic toxin concentration, and/or different dialysis vintage and techniques may explain differences when patients were stratified by sex and dialysis duration, respectively. Attenuated results in our cohort-specific analysis may be explained by differences in line of therapy in respective countries. Based on the above findings, sex, dialysis duration and/or regional differences in eating habits should be interrogated in more detail in future studies investigating links between lipid profile and uremic toxins. As eGFR is not suitable for dialysis patients without residual kidney function, we have further recalculated the multivariate models without including eGFR, where we found similar, but slightly increased, standardized beta values, further supporting our hypothesis of a negative association between uremic toxins and lipid profile.

Limitations of the present study include involvement of solely European centers in an observational, cross-sectional fashion, and the formula calculation of LDL cholesterol. Unfortunately, no dietary information, data on dialysis modality and performance, or outcome data are available for the current cohorts. We also acknowledge that adjustment for dialysis vintage and eGFR may not be sufficient for dialysis patients considering the effect of renal placement therapy on circulating toxins. We are, however, confident in our findings due to the minimal differences in regression models when patients were stratified by dialysis duration. Lastly, no information on genetic risk variants for an altered lipid profile are available in our cohort, considering the growing body of evidence that points to genetic determinants influencing different lipid particles [38]. On the other hand, our study presents many strengths, such as the analysis of a full lipid profile comprising of six clinically well-established lipid markers and a pathogenic uremic toxin profile consisting of five clinically relevant and centrally measured retention solutes. The use of three independent cohorts, access to clinical data to adjust for known risk factors, and a relevant number of patients within all three subcohorts further strengthen the results of our study.

In conclusion, the inverse associations between lipid profile and uremic toxins in KF highlight the complexity of the uremic environment. Our data suggest that not all uremic toxin interactions with conventional CV risk markers may be pathogenic.

## 4. Materials and Methods

### 4.1. Patients and Study Design

Briefly, the present study focuses on three cross-sectional KF cohorts from three centers across Europe: Stockholm, Sweden (N = 235); Leuven, Belgium (N = 150); and Leipzig, Germany (N = 226), with a total of 611 patients. The Stockholm subcohort is based on an ongoing study [39], with all adult patients undergoing living donor kidney transplantation (KTx) at the Department of Transplantation Surgery at the Karolinska University Hospital invited to participate in the study. All participants provided written informed consent and the Regional Ethical Review Board in Stockholm approved the study. All study participants were sampled immediately before KTx. Patients recruited from Leuven, Belgium, are part of a prospective, observational study on the natural evolution of CKD mineral bone disorder (CKD-MBD) after KTx (clinical trial identifier: NCT01886950). Patients included were referred for KTx from a deceased donor at the University Hospitals Leuven between October 2010 and April 2016. Use of anti-osteoporotic medications was the only exclusion criterion. Participants from Leipzig, Germany, were derived from an ongoing study investigating metabolic risk factors in patients with CKD [40]. Inclusion criteria were subjects >18 years of age and written informed consent. Exclusion criteria were end-stage malignant diseases, pregnancy, acute generalized inflammation, acute infectious disease, and history of drug abuse. Although not recorded, the majority of patients in all three cohorts were Caucasian, estimated to be over 95%. In all three cohorts, the local ethics committee from each study center approved the respective study, which was performed in accordance with the Declaration of Helsinki. All subjects gave written informed consent before taking part in the study.

### 4.2. Biochemical Analysis/Clinical Parameters

For all patients and study centers, blood specimens were drawn routinely at the time of transplantation (Stockholm and Leuven subcohorts) or on study entry (Leipzig subcohort). Subjects in Stockholm and Leipzig cohorts were fasted at time of blood collection, while subjects in Leuven cohort were randomly sampled since patients underwent deceased donor transplantation and this type of surgery cannot be planned. Routine parameters in all three study centers include measurements of hsCRP, total cholesterol, HDL cholesterol, triglycerides, and creatinine, all of which were quantified at the respective center. In all patients receiving dialysis treatment, blood was obtained just before dialysis started.

Non-HDL cholesterol was calculated by subtracting HDL cholesterol from total cholesterol. LDL cholesterol was calculated by the equation: Total cholesterol [mmol/L] − HDL cholesterol [mmol/L] + (K × triglycerides [mmol/L]) where K = 0.46 if triglycerides ≤ 4.5 mmol/L and K = 0.37 if triglycerides > 4.5 mmol/L [41]. According to Tremblay and co-workers [42], LDL cholesterol was not to be derived by the above-mentioned equation when triglycerides were >9 mmol/L. Remnant cholesterol was calculated as total cholesterol minus HDL cholesterol minus LDL cholesterol [43]. In all individuals, eGFR was assessed according to the creatinine-based Chronic Kidney Disease Epidemiology Collaboration (CKD-EPI) equation [44].

We decided to measure total rather than free uremic toxin levels, as most of the previous publications on cardiovascular effects of uremic toxins were derived from total uremic toxin measurements [11,21,22,23]. Concentrations of total indoxyl sulphate, p-cresyl sulphate, IAA, TMAO, and PAG were centrally quantified by a previously described method [45]. In brief, serum samples were deproteinized with acetonitrile after addition of an internal standard (stable isotope labeled analogues) and then filtered over a 96-well Ostro plate (Waters, Zellik, Belgium). After drying with nitrogen and redissolving in MilliQ water, the samples were analyzed using an ultra-performance liquid chromatography—tandem mass spectrometry with positive electrospray ionization. We have previously shown that no differences in uremic toxin levels are detectable after up to 10× freeze/thaw cycles [45]. Since all samples were thawed a maximum of three times, our results were not influenced by freeze-thaw cycles.

Patients on dialysis treatment were classified as prevalent dialysis patients if they were on dialysis treatment for six or more months according to Chen et al. [46]. At the time of recruitment for our tri-national cohort, most of the available oral glucose-lowering treatment was contraindicated in patients with advanced CKD and KF, including metformin and sulfonylureas, but also sodium–glucose cotransporter 2 inhibitors or glucagon-like peptide 1 receptor agonists. Therefore, these modern pharmaceutical approaches did not influence the results of our study.

### 4.3. Statistical Analysis

Continuous variables were expressed as median with interquartile range (25th to 75th percentile). Differences between two groups according to sex, dialysis duration, dialysis status, and cohort were determined using non-parametric Mann–Whitney U test/Kruskal-Wallis test for continuous parameters or Chi-squared test for categorical variables. Univariate correlations were assessed using non-parametric Spearman’s rank-order method, where *p* values corrected according to Bonferroni adjustment were deemed significant <0.05. To identify independent associations between uremic toxins and lipid profile, multivariate linear regression models for uremic toxins that were significantly correlated to lipid markers in univariate analyses were used. Covariates in the multivariate models included age at study entry, sex, cohort, presence of diabetes, hsCRP, eGFR, statin use, and dialysis treatment. In all multivariate models, non-normally distributed variables were log10 transformed and missing values were handled using a pairwise exclusion process (missing values: total cholesterol = 4; HDL cholesterol = 4; non-HDL cholesterol = 4; LDL cholesterol = 6; remnant cholesterol = 6; triglycerides = 4; indoxyl sulphate = 21; *p*-cresyl sulphate = 21; TMAO = 21; PAG = 21; IAA = 138). Finally, sensitivity analyses were conducted for statistically significant full models, stratified by sex, cohort, dialysis duration, or dialysis treatment. Covariates for models in sensitivity analyses were adapted where deemed relevant. For group-wise comparisons and all multivariate analyses, *p* values < 0.05 were deemed significant.

## Figures and Tables

**Figure 1 toxins-14-00412-f001:**
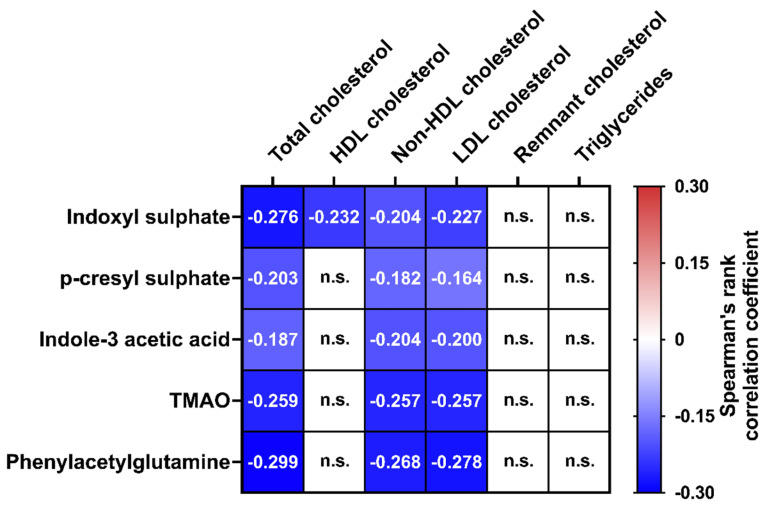
Heatmap of Spearman’s rank correlation coefficients for univariate correlations between the panel of five uremic toxins and six lipid parameters in the entire cohort (N = 611). Strict Bonferroni correction was applied for all univariate analyses and Bonferroni-corrected level of significance was *p* < 0.001666 (0.05/30 tests). Non-significant correlations are indicated by n.s., whereas significant associations are depicted as color-coded cells with exact r values inside. Thus, positive (in red/warmer colors) and negative associations (in blue/cooler colors) are shaded based on their respective Spearman’s rank correlation coefficients. Abbreviations: HDL, High density lipoprotein; LDL, Low density lipoprotein; TMAO, Trimethylamine-N-Oxide.

**Table 1 toxins-14-00412-t001:** Baseline characteristics of the entire study cohort (N = 611) and after stratification based on sex and dialysis status.

	Entire Cohort	Female Subjects	Male Subjects	*p*	Dialysis	Non-Dialysis	*p*
N	**611**	**212**	**399**	**-**	**423**	**188**	-
Age (years)	55 (43–67)	56 (44–68)	54 (42–65)	0.057	55 (43–65)	56 (42–70)	0.061
Male sex (N [%])	399 [65.3%]	-	-	-	287 [67.8%]	112 [59.6%]	**0.047**
BMI (kg/m^2^)	24.8 (22.6–27.5)	24.8 (22–28)	24.8 (22.9–27.4)	0.572	24.6 (22.5–27.8)	25.1 (23.1–27.3)	0.468
SBP (mmHg)	140 (126–153)	140 (125–154)	139 (126–153)	0.751	139 (121–153)	140 (129–152)	0.132
DBP (mmHg)	80 (72–90)	80 (71–90)	80 (72–90)	0.408	80 (70–90)	82 (75–90)	**0.029**
Diabetes mellitus (N [%])	53 [8.7%]	16 [7.5%]	37 [9.3%]	0.422	50 [12.0%]	3 [1.6%]	**<0.001**
Statin users (N [%])	231 [37.8%]	79 [37.3%]	152 [38.1%]	0.905	156 [39.8%]	75 [40.1%]	0.787
Dialysis (N [%])	423 [69.2%]	136 [64.2%]	287 [71.9%]	0.050	423 [100%]	0 [0%]	-
Vintage (months)	16 (3–43)	18 (5–45)	16 (3–43)	0.688	24 (11–47)	-	-
Creatinine (µmol/L)	637 (427–848)	541 (349–725)	700 (500–911)	**<0.001**	706 (556–901)	356 (172–631)	**<0.001**
eGFR (mL/min/1.73 m^2^)	7 (5–11)	7 (5–12)	7 (5–11)	0.658	6 (5–8)	13 (7–28)	**<0.001**
Albumin (g/L)	38 (33–42)	37 (33–42)	38 (34–42)	0.076	38 (34–42)	37 (22–40)	**0.040**
hsCRP (mg/L)	2.0 (0.9–4.9)	1.9 (0.9–5.9)	2.0 (0.8–4.4)	0.416	2.2 (1.0–5.9)	1.4 (0.7–3.4)	**<0.001**
Total chol. (mmol/L)	4.6 (3.9–5.5)	4.9 (4.2–6.2)	4.4 (3.7–5.2)	**<0.001**	4.5 (3.8–5.3)	4.9 (4.1–6.3)	**<0.001**
HDL chol. (mmol/L)	1.3 (1.0–1.7)	1.5 (1.2–1.8)	1.2 (1.0–1.5)	**<0.001**	1.3 (1.0–1.6)	1.4 (1.1–1.7)	**0.009**
Non-HDL chol (mmol/L)	3.2 (2.5–4.1)	3.4 (2.7–4.6)	3.2 (2.4–3.9)	**0.002**	3.1 (2.4–3.9)	3.6 (2.7–4.7)	**<0.001**
LDL chol. (mmol/L)	2.5 (1.8–3.3)	2.6 (2.0–3.7)	2.4 (1.8–3.1)	**<0.001**	2.3 (1.8–3.0)	2.8 (2.1–3.8)	**<0.001**
Remnant chol. (mmol/L)	0.7 (0.5–0.9)	0.7 (0.5–0.9)	0.7 (0.5–0.9)	0.592	0.7 (0.5–0.9)	0.7 (0.5–0.9)	0.675
Triglycerides (mmol/L)	1.4 (1.0–2.0)	1.4 (1.0–2.0)	1.5 (1.1–2.0)	0.630	1.4 (1.0–2.1)	1.5 (1.0–2.0)	0.749
Indoxyl sulphate (µmol/L)	87.8 (43.1–135.4)	80.5 (31.4–126.6)	92.8 (54.1–143.9)	**0.005**	106.4 (73.0–154.3)	27.9 (12.7–74.9)	**<0.001**
p-cresyl sulphate (µmol/L)	133.4 (68.0–202.7)	127.7 (69.2–203.0)	137 (67.2–202.3)	0.615	146.2 (85.8–207.6)	101.1 (45.4–187.1)	**<0.001**
Indole-3 acetic acid (µmol/L)	4.5 (3.0–6.3)	4.3 (2.7–5.9)	4.6 (3.1–6.6)	0.139	5.0 (3.4–7.1)	3.3 (2.4–4.6)	**<0.001**
TMAO (µmol/L)	57.6 (27.9–105.5)	54.6 (24.1–118.1)	60.2 (29.8–104.5)	0.514	71.4 (41.0–123.0)	25.9 (10.7–65.3)	**<0.001**
Phenylacetylglutamine (µmol/L)	46.1 (18.9–95.9)	45.6 (15.3–110.3)	46.5 (20.9–89.4)	0.936	71.5 (32.0–126.0)	13.0 (5.6–33.2)	**<0.001**

BMI, Body mass index; Chol, Cholesterol; DBP, Diastolic blood pressure; eGFR, estimated glomerular filtration rate; HDL, High density lipoprotein; hsCRP, High sensitivity C-reactive protein; LDL, Low density lipoprotein; SBP, Systolic blood pressure; TMAO, Trimethylamine-N-Oxide. Data are presented as median (interquartile range) for continuous measures, and N [percentage] for categorical measures. *p* values for differences between female and male subjects were assessed by non-parametric Mann–Whitney U test for continuous parameters or Chi-squared test for categorical variables, and significant *p* values (*p* < 0.05) are depicted in **bold**.

**Table 2 toxins-14-00412-t002:** Multiple linear regression analyses in the entire cohort (N = 611) between five uremic toxins (dependent variable) and lipid parameters adjusted for age, sex, study center, presence of diabetes, body mass index, high sensitivity C-reactive protein, estimated glomerular filtration rate, statin usage, and dialysis treatment (yes/no).

Analytes (µmol/L)		Total Cholesterol (mmol/L)	HDL Cholesterol (mmol/L)	Non-HDL Cholesterol (mmol/L)	LDL Cholesterol (mmol/L)	Remnant Cholesterol (mmol/L)	Triglycerides (mmol/L)
Indoxyl sulphate	**ꞵ**	−0.041	−0.009	−0.024	−0.018	-	-
	** *p* **	0.164	0.742	0.404	0.551	-	-
p-cresyl sulphate	**ꞵ**	−0.016	−	−0.007	0.038	-	-
	** *p* **	0.747	−	0.885	0.448	-	-
Indole-3 acetic acid	**ꞵ**	−0.091	−	**−0.150**	**−0.122**	-	-
	** *p* **	0.090	−	**0.004**	**0.025**	-	-
TMAO	**ꞵ**	−0.051	−	**−0.079**	−0.045	-	-
	** *p* **	0.165	−	**0.031**	0.229	-	-
Phenylacetylglutamine	**ꞵ**	**−0.095**	−	**−0.133**	**−0.071**	-	-
	** *p* **	**<0.001**	−	**<0.001**	**0.016**	-	-

A multivariate model was calculated only for those uremic toxins for which a Bonferroni-adjusted significant univariate correlation was found (Figure 1). Non-normally distributed variables were log10 transformed prior to analysis. Standardized ꞵ coefficients, as well as the respective *p*-values, are given for each model. Significant associations (*p* < 0.05) after adjustment for covariates are depicted in **bold**. Abbreviations are indicated in Table 1.

## Data Availability

The data underlying this article cannot be shared publicly due to ethical reasons, e.g., for the privacy of individuals that participated in the study. The data will be shared on reasonable request to the corresponding author.

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
