# Peer review of "Lipid Profile Is Negatively Associated with Uremic Toxins in Patients with Kidney Failure—A Tri-National Cohort"

_toxins, 2022, doi:10.3390/toxins14060412_

Round 1

Reviewer 1 Report

In this work “Lipid profile is negatively associated with uremic toxins in patients with kidney failure – a tri-national cohort”, the authors propose a work where mostly they highlight the pathogenicity of uremic toxin interactions with conventional CV risk marker as the lipid profile. The work is enough complete, but it is enough lacking on molecular mechanism and/or pathways that could explain the interaction between lipid profile and UTs (it is necessary to be hint only). Also, it should be emphasized more the the metabolomic aspect and their role in pathophysiology https://www.ncbi.nlm.nih.gov/pmc/articles/PMC5198570/  https://pubmed.ncbi.nlm.nih.gov/24072415/. It is important to define whether if there is a genetic independent alteration of lipid metabolism within the selected cohort. If not, it must be specified that it has not been done in the paper. https://www.nature.com/articles/nrg2321

Reviewer 2 Report

The authors examined associations between uremic toxins and lipid profiles in kidney disease patients in three regions. It may be important to understand the associations for the strategies of CKD-induced systemic disorders.

1.The authors should put tables and figures in the manuscript.

  1. In Table 1, to present eGFR is not suitable for dialysis patients with no residual kidney functions.
  2. The authors showed the association between uremic toxins and lipid profiles was different among the regions, and discussed the reasons, such as eating habit, sex, dialysis duration, and so on. It is interesting that association between PCS and TC was different completely in Leuven compared with other regions, and the authors should show patient background in each region.

Reviewer 3 Report

The aim of this study is to determine the relation between 5 main uremic toxins and lipids profils in a large cohort of CKD patients and hemodialysis patients.

The strength of this study are:

1- the number of patients and the multicentric design and  the subanalysis for each center

2-The statsitcal anlaysis were well done

3_ the manuscript is clear and well written

I have several concerns:

1° The table 1 and 2 were not in the manuscript.

2° Please specify if patients were fasted at the time of blood collection. Please precise how the samples were storred and how many freeze and defreeze. All these parameters could influence the analyses. 

3°In the multivariate analysis the BMI were not included. Also the diabetes treatment ( ex metformin) could influence the results and probably must be a critere of adjustment;

4° We have no information at all about the diet that could influence uremic toxins and lipids profiles. The authors have already discussed this point in the limitation.

5° Does some patients were treated with ezetimibe or similar molecules ?

6° Why the same analysis have been not performed with free uremic toxins ? In particular we have no information about albumin level? 

Also what is the dialysis technic and performance ( KT/V ? HDF ? ). These points should at least be discussed. 

Round 2

Reviewer 2 Report

The authors have revised their manuscript according to the suggestions from the reviewers.